# Compute Or Load KV Cache? Why not Both?

Shuowei Jin [* 1]   Xueshen Liu [* 1]   Qingzhao Zhang [1]   Z. Morley Mao [1]

## Abstract

Large Language Models (LLMs) are increasingly deployed in large-scale online services, enabling sophisticated applications. However, the computational overhead of generating key-value (KV) caches in the prefill stage presents a major bottleneck, particularly for long-context inputs. Prefix caching mitigates this issue by storing KV caches for reuse, reducing redundant computation. Despite its advantages, prefix caching suffers from high latency due to the limited I/O bandwidth of storage devices, constraining inference efficiency. To address this challenge, we introduce *Cake*, a novel KV cache loading system that optimally utilizes both computational and I/O resources in parallel. *Cake* employs a bidirectional scheduling strategy that dynamically balances KV cache computation and loading, ensuring efficient resource utilization. Additionally, *Cake* incorporates an adaptive scheduling mechanism that seamlessly integrates with non-prefix caching requests, improving system throughput and adapting to fluctuating resource availability. Through extensive evaluations across various hardware configurations, datasets, and storage conditions, *Cake* achieves on average **2.6×** reduction in Time to First Token (TTFT) compared to compute-only and I/O-only methods. Our findings highlight *Cake* as an effective and practical solution for optimizing long-context LLM inference, bridging the gap between computation and I/O efficiency in large-scale AI deployments.

## 1. Introduction

Large Language Models (LLMs) have become a cornerstone of modern AI applications, powering a wide range of large-scale online services. As these models are increasingly adopted, ensuring efficient online inference has emerged as a critical research and engineering challenge (Agrawal et al., 2024; Zheng et al., 2023; Miao et al., 2024; Leviathan et al., 2023; Ning et al., 2023; Jin et al., 2025). Recent advancements in LLMs, such as the expansion of context windows (openAI, 2024; Anthropic, 2024), have enabled sophisticated applications including long document understanding (Wang et al., 2024), long-context Retrieval-Augmented Generation (RAG) (Jiang et al., 2024), and the creation of complex LLM-based agents (Zhang et al., 2024). However, although these capabilities improve utility, they also introduce significant computational overhead during inference, due to the cost of generating key–value (KV) caches.[1] The resulting latency can noticeably degrade user experience. Given this computational bottleneck, strategies to optimize LLM inference workflows are essential.

In real-world applications, many tokens are reused across users and conversations, presenting an opportunity for system optimization. For instance, in multi-turn conversations, follow-up queries reuse the key-value (KV cache) pairs from prior tokens, while in Retrieval-Augmented Generation (RAG) workflows, document KV cache can be shared across multiple user queries (Jin et al., 2024; Chan et al., 2024). To reduce these redundancies, **prefix caching** stores previously computed KV cache and loads them into GPU memory for inference when the corresponding tokens are hit. Leading LLM service providers, including OpenAI, Anthropic, and DeepSeek, have integrated prefix caching mechanisms into their inference systems, lowering inference costs by over 50% (OpenAI, 2024; Anthropic, 2023; Deepseek, 2024).

Despite its advantages, deploying prefix caching at scale poses a key challenge: designing a high-capacity KV cache management system across heterogeneous memory layers while ensuring low loading latency for optimal **Time-to-First-Token (TTFT)** during inference. State-of-the-art inference engines (Kwon et al., 2023; Zheng et al., 2023) typically store KV cache in GPU and CPU memory, ensur-

---

*Equal contribution   [1]University of Michigan. Correspondence to: Shuowei Jin <jinsw@umich.edu>, Xueshen Liu <liuxs@umich.edu>.

*Proceedings of the 42nd International Conference on Machine Learning*, Vancouver, Canada. PMLR 267, 2025. Copyright 2025 by the author(s).

---

[1]KV cache is widely used in state-of-the-art inference systems to reduce computational overhead during decoding for each request.

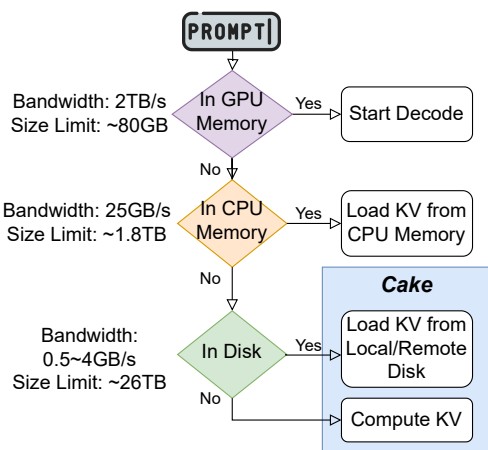

*Figure 1.* Workflow of long-context LLM inference with prefix caching. *Cake* operates during the KV cache loading phase (highlighted in blue). The configuration parameters are based on the specifications of a LambdaLab GPU server (Lambda, 2024).

ing minimal latency. However, GPU and CPU memory are both limited and expensive, often requiring the eviction of KV cache for long-context requests, making them impractical for large-scale inference services. To address this limitation, hierarchical storage systems have been proposed, leveraging CPU memory, local disks, and remote storage for managing KV cache caches (Gao et al., 2024; Liu et al., 2023; Yao et al., 2024). Figure 1 illustrates the hierarchical workflow of LLM inference with prefix caching. Upon receiving a query, the system searches for reusable KV cache caches across three storage tiers:

- **GPU Memory**: The fastest but most capacity-limited option (bandwidth: ∼2TB/s, size: ∼80GB). If the KV cache cache is found, decoding proceeds immediately.
- **CPU Memory**: Offers greater capacity (bandwidth: ∼25GB/s, size: ∼1.8TB) but is slower than GPU memory.
- **Disk Storage**: The largest capacity option (bandwidth: 0.5–4GB/s, size: ∼26TB) but also the lowest bandwidth.
- **Compute**: If no KV cache cache is found across all levels, the system computes it from scratch.

Due to the substantial capacity differences in the storage hierarchy, a significant portion of KV cache caches reside on local or remote disks. As evaluated in AttentionStore (Gao et al., 2024), approximately 80% of cache hits occur at the disk level. However, fetching large KV cache from disk is constrained by the low I/O bandwidth of PCIe or network-based remote storage, significantly impacting TTFT, which adversely affects user experience.

**Contribution.** In this paper, we introduce *Cake* (Computation and I/O Aware KV cache CachE loader), a

novel KV cache loading system designed to minimize latency when loading KV cache from high-capacity, low-bandwidth storage layers. *Cake* optimally leverages the distinct characteristics of both computational and I/O resources in parallel, significantly reducing TTFT by 2.6x on average in long-context prefix caching scenarios.

In addition to reducing latency, *Cake* is designed considering real-world practicality. We propose an adaptive scheduling mechanism that efficiently shares compute resources with non-prefix caching requests, improving overall throughput by 26%. Additionally, *Cake* dynamically adjusts to fluctuations in network and computational resources, ensuring consistently optimal latency.

Furthermore, we conduct extensive experiments across diverse setups and provide a detailed analysis of *Cake*'s effectiveness. Our findings offer practical insights into when *Cake* achieves the greatest performance gains, serving as a valuable guide for future real-world deployments.

To the best of our knowledge, *Cake* is the first system to demonstrate that efficiently utilizing both computational and I/O resources can optimally reduce TTFT in long-context prefix caching scenarios. Prior approaches either rely exclusively on computation or solely on I/O for KV cache loading (Liu et al., 2023; Kwon et al., 2023), leaving a significant gap in understanding how to leverage the unique characteristics of both computation and I/O for efficient KV cache cache loading. *Cake* addresses this gap, providing a comprehensive and adaptive solution for long-context prefix caching scenarios.

## 2. Related Work

Previous work has explored two primary directions to develop practical prefix caching systems: (1) algorithm-level compression techniques that reduce the size of KV cache, thereby decreasing loading time, and (2) system-level KV cache management strategies that expand cache capacity across heterogeneous memory layers.

**KV cache Compression.** Most work compress the KV cache through quantization, token pruning, and model architectural modifications. Quantization methods (Hooper et al., 2024; Kang et al., 2024; Liu et al., 2024b) reduce the precision of KV cache representations while maintaining accuracy. Token pruning approaches like LLMLingua (Jiang et al., 2023), ScissorHands (Liu et al., 2024a), and H2O (Zhang et al., 2023) identify and remove less important tokens from the KV cache. At the model architecture level, Grouped-Query Attention (GQA) (Ainslie et al., 2023) reduce memory footprint by sharing key-value heads across queries in the group, while Multi-head Latent Attention (MLA) compresses KV cache into compact latent vectors to reduce KV cache size.

**System Optimizations.** vLLM (Kwon et al., 2023) and SGLang (Zheng et al., 2023) mainly optimize KV cache management between GPU DRAM and CPU memory, enabling low-latency loading despite limited capacity. CachedAttention (Gao et al., 2024) extends this by employing a hierarchical KV cache management system across memory and disk mediums, effectively support multi-turn conversations. CacheGen (Liu et al., 2023) addresses scenarios where KV cache is stored in remote data storages, applying adaptive compression methods to reduce the latency of loading through network.

While these approaches focus on reducing the I/O load latency to reduce TTFT, our work tackles the problem from a different angle: combining computation and I/O in parallel to further reduce latency. It is an orthogonal design to existing methods.

## 3. Background

**Attention and KV cache.** The attention mechanism (Vaswani et al., 2017) is a core component of LLMs, allowing them to model token relationships efficiently. KV cache is a technique to improve attention module inference efficiency.

Given an input sequence $X$, the attention module first transforms it into queries, keys, and values:

$$Q = XW_Q, \quad K = XW_K, \quad V = XW_V$$

where $W_Q$, $W_K$, and $W_V$ are learned projection matrices. The attention scores are then computed as:

$$\text{Attention}(Q, K, V) = \text{softmax}\left(\frac{QK^T}{\sqrt{d_k}}\right)V$$

where $d_k$ is the key dimension.

During autoregressive decoding, this computation repeats at each step $t$, generating a new token $x_t$ based on previous tokens $x_{<t}$. However, the key and value vectors for $x_{<t}$ remain unchanged across decoding steps. To eliminate redundant computation, modern systems cache these as past_K and past_V, computing only the query for $x_t$ along with the new key $k_t$ and value $v_t$. This optimization, known as KV cache, significantly reduces computational overhead.

**Chunk Prefill.** Chunk prefill is a technique used to optimize LLM inference, particularly for long input sequences. State-of-the-art LLM inference engines (Kwon et al., 2023; Zheng et al., 2023) split inference into two phases:

- **Prefill:** Computes the KV cache for the input prompt.
- **Decode:** Generates tokens sequentially using the cached KV, updating it incrementally.

For long-context inputs, the prefill stage is highly compute-intensive, requiring substantial GPU resources for a long period. Chunk prefill (Agrawal et al., 2023; 2024) mitigates this by dividing the input sequence into smaller chunks to prefill it chunk by chunk and batching small chunk prefill together with decode requests. To be specific, vLLM forms a batch of requests for each inference step based on a predetermined token budget. The scheduler prioritizes decode requests, allocating one token from the budget to each. Any remaining tokens in the budget are then assigned to prefill requests for chunk prefill. This prevents long prefill operations from blocking decode requests, improving GPU utilization by co-locating compute-bound (prefill) and memory-bound (decode) tasks in the same batch. Currently, it is the default suggested scheduling mode in vLLM (Kwon et al., 2023).

We design *Cake* upon this approach, scheduling chunks for either computation or I/O. Compared to token-level scheduling, chunk-level scheduling better exploits GPU parallelism for maximum efficiency. Compared to sequence-level scheduling, it is more fine-grained, allowing greater flexibility for optimal scheduling strategies.

## 4. Design of *Cake*

We design *Cake* to optimize the latency of loading KV cache from storage layers with limited bandwidth but large capacity, such as local disks and remote storage —where most cache hits occur in prefix caching scenarios. *Cake* leverages both compute and I/O resources bidirectionally and in parallel to minimize this latency. We present *Cake* workflow in Figure 2.

In this section, we first analyze the compute and I/O capability in common inference server setups. Next, we present key insights into *Cake* 's scheduling design, focusing on how to efficiently utilize the unique characteristics of compute and I/O. Finally, we discuss in real-world scenarios how *Cake* dynamically adapts to fluctuations in compute and I/O bandwidth, ensuring optimal latency.

**Part 1: Analysis of Compute and I/O Capability.** We first analyze the performance characteristics of KV cache loading and computation across common inference system configurations. We evaluate the equivalent throughput (calculated as KV cache file size divided by processing time) using vLLM (Kwon et al., 2023) with a chunk size of 512 tokens on the LongAlpaca-7B model (Chen et al., 2023). Our experiments use various GPUs to process a random context of 32k tokens. A more detailed analysis across various configurations is provided in Section 5.

Our results in Figure 3 demonstrate that computation and I/O resources achieve comparable throughput in typical inference scenarios. Specifically, computing KV cache with

| | Compute | | Unfinished Chunks | | | | Loaded Chunks | | I/O Load | | | |
|---|---|---|---|---|---|---|---|---|---|---|---|---|
| **Stage0:** | E | ating | cake | makes | me | happy | , | I | enjoy | eating | cakes | . |
| **Stage1:** | E | ating | cake | makes | me | happy | , | I | enjoy | eating | cakes | . |
| .... | | | | | | | | | | | | |
| **StageN-1:** | E | ating | cake | makes | me | happy | , | I | enjoy | eating | cakes | . |
| **StageN** | E | ating | cake | makes | me | happy | , | I | enjoy | eating | cakes | . |

Figure 2. Workflow of *Cake*: Computation starts from the beginning of the sequence, while I/O loading starts from the end. Both processes progress in parallel and merge in the middle, ensuring efficient KV cache loading and minimal latency.

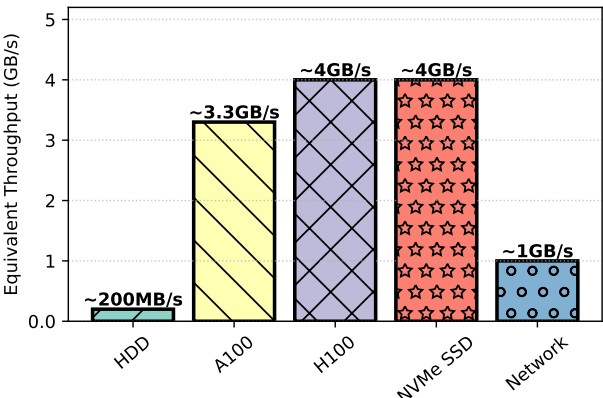

Figure 3. Comparison of equivalent KV cache loading bandwidth (bytes/second) across different storage mediums and GPU computation. (Bandwidth for GPU computation is calculated by dividing the total KV cache size by processing time.)

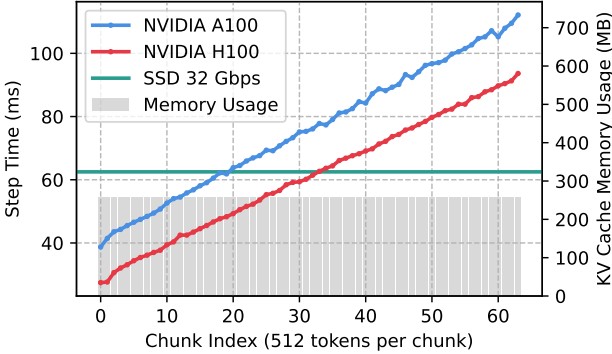

Figure 4. Chunk prefill time per step using different methods v.s. chunk index.

an H100 GPU achieves similar throughput to loading from SSD and is much faster than network bandwidth used in Google Cloud Egress (~1GB/s) (Google Cloud, 2023). This observation validates our design principle of leveraging both compute and I/O resources in parallel to minimize latency.

**Part 2: Scheduling Based on the Distinct Characteristics of Compute and I/O.** We design the *Cake* core scheduler as a bidirectional KV cache loader based on the following key insight:

> **Insight:** Compute cost increases for later tokens, while I/O cost remains constant regardless of token position.

This is because, in the attention operation, later tokens must attend to all previous tokens, resulting in a higher computational cost as the sequence progresses. However, the size of key-value vectors remain the same regardless of position, meaning I/O access cost is uniform across all tokens.

To validate this hypothesis, we conducted long-context prefill experiments using chunked prefill, where each chunk contains a fixed 512 tokens. As shown in Figure 4, the results exhibit a clear pattern: latency per chunk increases linearly with its index, with earlier chunks requiring less computation, while KV cache memory usage remains constant across all chunks.

Leveraging this insight, *Cake* schedules earlier chunks for computation and later chunks for I/O loading. As illustrated in Figure 2, upon receiving a request, *Cake* splits the sequence into chunks and initiates two parallel processes: (1) The local GPU computes KV cache from the beginning chunk, progressing forward. (2) The data loading process fetches KV cache from the last chunk, moving in reverse direction. This bidirectional strategy continues until the two processes meet in the middle, completing KV cache generation for the entire prompt.

**Part 3: Practical Considerations for Real-World Deployment.** In practical deployment, *Cake* is designed to address two key challenges: (1) Efficiently sharing compute resources with non-prefix caching requests. (2) Handling fluctuations in compute and I/O availability.

For the first challenge, real-world inference scenarios include requests without saved prefix cache that require compute resources for prefill and decoding. To ensure system throughput, *Cake* can prioritize other users' compute needs, allocating only available resources to *Cake* computation. Since *Cake* can still rely on I/O loading even when compute resources are constrained, we introduce an **adaptive scheduling mode** that extends vLLM's scheduling logic as discussed in Section 3. This mode follows a prioritized allocation order: 1. Decoding requests. 2. Non-prefix-cache chunked prefill. 3. Prefix-cache chunked prefill.

For the second challenge, since compute resources are shared across multiple users, and I/O bandwidth could fluctuate over time, *Cake* must dynamically adapt to these variations while maintaining optimal latency. However, this challenge is naturally mitigated by *Cake*'s bidirectional parallel design—whenever I/O or compute availability changes, the merging point of the two processes shifts accordingly, always ensuring minimal latency.

## 5. Evaluation

In this section, we evaluate *Cake* across a variety of real-world setups and discuss its benefits and limitations in deployment. These insights can guide future implementations and optimizations. We begin by outlining the experiment setup and then conduct comprehensive experiments to assess the impact of the following factors on *Cake*:

1. Varying I/O bandwidth and compute configurations (Section 5.2). 2. Sequence length (Section 5.4). 3. Model architecture (Section 5.5). 4. KV cache compression techniques (Section 5.6). 5. Resource fluctuation scenarios (Section 5.7). 6. Adaptive Scheduling (Section 5.8). 7. System overhead (Section 5.9).

### 5.1. Experiment Setup

**Models.** We evaluate *Cake* on various long-context models with different architectures and sizes, including LongAlpaca-7B and LongAlpaca-13B (Chen et al., 2023), which are based on LLaMA 2, as well as LLaMA 3.1-8B and LLaMA 3.1-70B. Due to hardware constraints, we use the FP8-weight version for LLaMA 3.1-70B, while all other models use FP16 weights. The first two are multi-head attention (MHA) models, while the last two apply group query attention (GQA), which will introduce different computation vs. memory. We use BF16 as the default kv cache data type. The details of the model are listed in Table 1.

**Evaluation Metrics.** We use time-to-first-token (TTFT) as our primary evaluation metric. TTFT is widely used in LLM inference, measuring the time between the arrival of

| Model | #Layers | KV-size/Token | #KV Hd | #Attn Hd |
|---|---|---|---|---|
| LongAlpaca-7B | 32 | 512 kB | 32 | 32 |
| LongAlpaca-13B | 40 | 800 kB | 32 | 32 |
| LLaMA 3.1-8B | 32 | 128 kB | 8 | 32 |
| LLaMA 3.1-70B | 80 | 320 kB | 8 | 64 |

*Table 1.* Comparison of experimental model configs.

a user query and the generation of the first token. In other words, it reflects either the time of loading stored KV cache or computing new KV cache.

**Datasets.** We evaluate *Cake* across various context lengths using three datasets with different task types: LongChat (Li et al., 2023) for multi-turn conversations, and TriviaQA and NarrativeQA (Bai et al., 2023) for long-document question-answering tasks. According to statistical analysis from CacheGen (Liu et al., 2023), most dataset queries range between 4k and 16k tokens in length. Since specific token values do not impact *Cake*'s performance evaluation (only token length matters), we generate synthetic prompts by uniformly sampling token lengths every 2k tokens within this range. Additionally, as discussed, *Cake* is designed to optimize the latency of loading **cached** KV cache from high-capacity, low-bandwidth storage layers. Thus, throughout our evaluation, we precompute and store all requests' KV cache in advance.

**Baselines.** We compare *Cake* to two types of KV cache prefill/loading mechanisms:

• Compute-only methods, which employ chunk prefill to compute all KV cache. We use vLLM (v0.6.2) in chunk prefill mode with token budget sizes of 512 by default. In Section 5.3, we evaluate how different token budget sizes affect Cake's performance.

• I/O Load-only, which loads saved KV cache from local/remote disks through disk/network. We use LMCache (v0.1.4).

**Hardware setting.** We run our evaluation on two server configurations: 1) A server equipped with two NVIDIA A100 80GB GPUs connected via NVLink, a 64-core AMD EPYC 7763 CPU, and 2.0TB of memory. 2) A server with a single NVIDIA H100 GPU, a 26-core vCPU, and 200GB of memory.

**I/O Bandwidth Control.** To accurately manage I/O bandwidth and ensure reproducibility, We simulate the chunk I/O loading process by calculating the appropriate delay time based on the chunk size and network bandwidth. The simulated storage backend is then set to pause data transfer to GPU memory until the specified delay has elapsed. We choose a variety of bandwidth configurations as demonstrated in Table 2. We choose the I/O loading part chunk size as 128 tokens as it empirically strikes an optimal bal-

| Physical Config | Bandwidth |
|---|---|
| Google Cloud standard egress limit | 7 Gbps |
| Google Cloud tier-1 egress limit | 25 Gbps |
| Lambda Lab SSD read | 32 Gbps |
| Samsung 980 pro SSD read | 56 Gbps |
| Infiniband (RoCE) | 100 Gbps |

*Table 2.* I/O bandwidth with corresponding physical configuration

| Hardware | Util | 7 Gbps | 25 Gbps | 32 Gbps | 56 Gbps | 100 Gbps |
|---|---|---|---|---|---|---|
| 2xA100 | 12.5% | 1.87\2.18 | 1.25\5.06 | 1.18\6.12 | 1.09\9.54 | 1.04\15.27 |
| 2xA100 | 50% | 4.57\1.29 | 2.07\2.03 | 1.80\2.24 | 1.46\3.06 | 1.30\4.62 |
| 2xA100 | 87.5% | 7.22\1.15 | 2.75\1.53 | 2.36\1.67 | 1.83\2.17 | 1.50\3.02 |
| 2xA100 | 100% | 8.10\1.10 | 3.03\1.43 | 2.63\1.59 | 1.97\2.00 | 1.54\2.64 |
| A100 | 12.5% | 1.57\2.85 | 1.14\7.22 | 1.10\8.95 | 1.03\14.06 | 1.02\23.59 |
| A100 | 50% | 3.35\1.46 | 1.68\2.55 | 1.55\3.01 | 1.30\4.26 | 1.17\6.48 |
| A100 | 87.5% | 5.05\1.26 | 2.18\1.88 | 1.90\2.10 | 1.56\2.91 | 1.30\4.07 |
| A100 | 100% | 5.62\1.21 | 2.41\1.81 | 2.00\1.91 | 1.69\2.71 | 1.30\3.52 |
| H100 | 12.5% | 1.74\2.40 | 1.21\5.80 | 1.13\6.90 | 1.08\11.08 | 1.05\18.24 |
| H100 | 50% | 4.13\1.38 | 1.94\2.25 | 1.72\2.55 | 1.45\3.62 | 1.27\5.34 |
| H100 | 87.5% | 6.23\1.17 | 2.53\1.66 | 2.23\1.86 | 1.74\2.44 | 1.50\3.54 |
| H100 | 100% | 7.13\1.17 | 2.74\1.56 | 2.40\1.75 | 1.84\2.25 | 1.49\3.08 |

*Table 3.* Speedup over I/O-only \ compute-only methods across different compute and I/O configurations. Chunk size: 512, Model: LongAlpaca-13B, Sequence length: 16K.

ance between bandwidth utilization and processing granularity.

**GPU Resource Utilization.** In online serving scenarios, a single machine often handles multiple user requests concurrently. As a result, a user's prefill operation may not always have access to the full GPU resources. To evaluate different GPU resource availability conditions, we adopt vLLM's token budget scheduling policy, as discussed in §3, to represent resource utilization. For instance, if the total token budget is 512 tokens and a *Cake* request consumes 256 tokens while the remaining tokens are allocated to other users, we define this scenario as 50% GPU utilization. We use this definition to simulate varying levels of GPU resource utilization.

**KV cache Compression methods.** KV cache Compression methods, which are orthogonal to our work. They can compress the KV cache size to make them more efficiently transferable through I/O. In our evaluation, we combine the most common 8-bit quantization and 3-bit quantization proposed by KVQuant (Hooper et al., 2024) with *Cake* to further evaluate its performance.

### 5.2. Evaluation Across Compute and I/O Configurations

In this section, we evaluate the performance of *Cake* under varying computational resources and I/O settings. Table 3 presents the speedup achieved by *Cake* compared to an I/O-only approach and a compute-only approach across different GPU hardware, GPU utilization levels, and I/O configurations. We conduct tensor parallel inference using a 2×A100 setup.

We observe that, for a fixed GPU utilization level, the speedup over I/O-only increases progressively from A100 to H100 and further to 2×A100. For example, under 100% GPU utilization with an I/O load bandwidth of 32 Gbps (representing a common server SSD read bandwidth), the speedup over I/O-only improves from 2× on A100 to 2.23× on H100, and 2.63× on 2×A100. This aligns with our design expectation that as compute resources increase, the performance advantage over an I/O-only approach continues to grow. Similarly, analyzing different GPU utilization levels within the same hardware setup yields the same

trend. Additionally, examining Table 3 row by row, we observe that as I/O bandwidth increases, the speedup over compute-only also improves. This highlights that *Cake* effectively balances both compute and I/O resources to enhance overall performance.

From a broader perspective, the most favorable deployment scenario for *Cake* occurs when its speedup is approximately 2× compared to both compute-only and I/O-only approaches. This scenario arises when computation and I/O capabilities are well-balanced, enabling *Cake* to efficiently utilize both in parallel and achieve the best average speedup over either baseline. We observe that in most cases, *Cake* provides a significant improvement, demonstrating its strong performance.

To aid interpretation, we highlight scenarios with at least a 1.5× improvement over both baselines in teal, indicating cases where *Cake* effectively leverages both resources. Conversely, we use text in red to mark cases where *Cake* achieves more than a 10× improvement over one baseline, which suggests a severe bottleneck in the other resource. In such extreme cases, comparing with the more capable resources, *Cake*'s speedup is limited.

To ensure a fair evaluation of *Cake* 's average performance improvement over the two baseline methods, we exclude the data points highlighted in red, as comparing *Cake* 's speedup relative to an extremely weak baseline would be misleading. After filtering these outliers, we compute the average speedup to provide a more balanced assessment of *Cake* 's overall efficiency. On average, *Cake* achieves 2.23x and 3.76x speedup over I/O-only and compute-only methods respectively, under different bandwidth and computation utilization levels.

### 5.3. Evaluation Across Varying Chunk Sizes

In this section, we evaluate the performance of *Cake* under varying chunk sizes configurations (i.e., the number of batched tokens) in vLLM's chunked prefill mode.

| ChunkSize | BW | LongAlpaca-7B | LongAlpaca-13B |
|---|---|---|---|
| 64 | 32Gbps | 1.47\3.47 | 1.47\3.55 |
| 128 | 32Gbps | 1.83\2.40 | 1.79\2.45 |
| 256 | 32Gbps | 2.06\2.09 | 1.96\2.20 |
| 512 | 32Gbps | 2.20\1.94 | 2.03\1.91 |
| 1024 | 32Gbps | 2.15\1.70 | 2.18\1.91 |
| 2048 | 32Gbps | 2.31\1.71 | 2.04\1.70 |

*Table 4.* Speedup over I/O-only \ compute-only methods across different chunk sizes of tokens and different models. Hardware 1xA100, Utilization 100%, Sequence length: 16K.

As shown in Table 4, *Cake* achieves an average Time-to-First-Token (TTFT) speedup of 1.96× over I/O-only methods and 2.25× over compute-only methods across different chunk sizes. Smaller chunk sizes lead to underutilization of computational resources, while larger chunk sizes enable more efficient utilization. In general, the speedup over I/O-only methods increases with chunk size, reflecting *Cake*'s growing reliance on computation. These results demonstrate *Cake*'s adaptability to varying compute efficiencies induced by different chunk configurations, allowing it to automatically optimize TTFT.

### 5.4. Evaluation Across Sequence Lengths

In this section, we evaluate the performance of *Cake* under varying sequence length settings. Table 5 presents the speedup achieved by *Cake* compared to an I/O-only approach and a compute-only approach across different compute-I/O configurations and sequence length settings.

We observe that, for a fixed computation and I/O configuration, increasing the sequence length consistently improves the speedup over compute-only methods. As discussed in Section 4, later tokens in a sequence require more computation. Therefore, as sequence length grows, leveraging I/O-only methods to load tokens from later positions becomes increasingly beneficial compared to compute-only approaches, leading to a continuous increase in speedup.

Following the interpretation aid strategy in Section 5.2, We highlight less effective scenarios in red and highly beneficial scenarios in teal, observing that *Cake* is highly beneficial in many cases. Notably, there is a single scenario where *Cake* underperforms compared to compute-only methods. This occurs when the sequence is short, as using 2×A100 for computation is significantly faster than loading data via a 7 Gbps bandwidth. However, due to *Cake* 's scheduling algorithm, it still assigns a portion of the workload to the I/O-only method, introducing overhead. For future work, we can optimize this scheduling strategy by incorporating an estimation mechanism. If one resource significantly outperforms the other, *Cake* could adaptively fall back to a single-resource mode, utilizing only the more efficient resource to minimize overhead.

However, on average, we can still observe *Cake* improves

| Hardware | BW | Util | 4k Tokens | 8k Tokens | 12k Tokens | 16k Tokens |
|---|---|---|---|---|---|---|
| 2xA100 | 7Gbps | 12.5% | 2.02\1.83 | 1.97\1.95 | 1.94\2.09 | 1.87\2.18 |
| 2xA100 | 7Gbps | 50% | 4.75\1.08 | 4.97\1.21 | 4.71\1.24 | 4.57\1.29 |
| 2xA100 | 7Gbps | 87.5% | 7.31\1.00 | 7.49\1.06 | 7.61\1.15 | 7.22\1.15 |
| 2xA100 | 7Gbps | 100% | 9.06\0.99 | 8.65\1.04 | 8.43\1.08 | 8.10\1.10 |
| 2xA100 | 32Gbps | 12.5% | 1.13\4.41 | 1.19\5.17 | 1.19\5.68 | 1.18\6.12 |
| 2xA100 | 32Gbps | 50% | 1.68\1.64 | 1.87\1.99 | 1.80\2.09 | 1.80\2.24 |
| 2xA100 | 32Gbps | 87.5% | 2.47\1.45 | 2.45\1.52 | 2.49\1.66 | 2.36\1.67 |
| 2xA100 | 32Gbps | 100% | 2.67\1.25 | 2.73\1.43 | 2.69\1.52 | 2.63\1.59 |
| 1xA100 | 7Gbps | 12.5% | 1.62\2.30 | 1.59\2.47 | 1.58\2.67 | 1.57\2.85 |
| 1xA100 | 7Gbps | 50% | 3.58\1.27 | 3.48\1.32 | 3.45\1.41 | 3.35\1.46 |
| 1xA100 | 7Gbps | 87.5% | 5.36\1.12 | 5.18\1.15 | 5.17\1.21 | 5.05\1.26 |
| 1xA100 | 7Gbps | 100% | 6.31\1.11 | 5.95\1.13 | 5.66\1.15 | 5.62\1.21 |
| 1xA100 | 32Gbps | 12.5% | 1.04\6.37 | 1.06\7.20 | 1.08\8.04 | 1.10\8.95 |
| 1xA100 | 32Gbps | 50% | 1.38\2.10 | 1.49\2.48 | 1.50\2.71 | 1.55\3.01 |
| 1xA100 | 32Gbps | 87.5% | 1.83\1.65 | 1.93\1.87 | 1.91\1.97 | 1.90\2.10 |
| 1xA100 | 32Gbps | 100% | 2.06\1.55 | 2.15\1.79 | 2.00\1.79 | 2.00\1.91 |

*Table 5.* Speedup over I/O-only \ compute-only methods across different sequence length settings, Hardware: 2xA100, Chunk size 512, Model: Long-Alpaca-13B

prefilling speed 3.34x faster than the I/O-only method and 2.24x faster than the compute-only method.

### 5.5. Evaluation Across Model Architectures

Table 6 presents the speedup achieved by *Cake* compared to an I/O-only approach and a compute-only approach across different compute-I/O configurations and model architectures. Both LongAlpaca models employ the Multi-Head Attention (MHA) mechanism, while the LLaMA 3 series utilizes Grouped Query Attention (GQA) to compress the KV cache, thereby reducing KV cache memory overhead.

By comparing the LongAlpaca-7B and LLaMA 3.1-8B models under the same compute and I/O settings, we observe that, despite the I/O bandwidth remaining unchanged, the speedup relative to the compute-only method increases. This is due to the benefits of GQA, which reduces the KV cache size, effectively enhancing I/O efficiency.

Furthermore, comparing different model sizes within the same series reveals that the speedup for I/O-only methods decreases as the model size grows. This indicates that computation requirements scale faster than I/O demands, leading to diminishing relative gains from compute optimizations.

We also identified four data points where *Cake* did not outperform the baseline, similar to the issue discussed in Section 5.4. This can be mitigated by incorporating a fall-back mechanism that dynamically adjusts resource allocation when one resource significantly outperforms the other. Despite the overheads on extreme scenarios, *Cake* on average achieves 2.68x speedup over the I/O-only method and 3.01 speedup over the compute-only method.

| Util | BW | LongAlpaca-7B | LongAlpaca-13B | Llama3.1-8B | Llama3.1-70B |
|------|------|------|------|------|------|
| 12.5% | 7Gbps | 1.92\2.15 | 1.87\2.18 | 1.20\5.56 | 0.99\12.92 |
| 12.5% | 32Gbps | 1.20\5.98 | 1.18\6.12 | 0.98\19.47 | 0.80\44.26 |
| 50% | 7Gbps | 4.79\1.30 | 4.57\1.29 | 1.88\2.13 | 1.25\3.99 |
| 50% | 32Gbps | 1.85\2.22 | 1.80\2.24 | 1.13\5.42 | 0.91\12.24 |
| 100% | 7Gbps | 8.55\1.10 | 8.10\1.10 | 2.81\1.52 | 1.66\2.64 |
| 100% | 32Gbps | 2.79\1.60 | 2.63\1.59 | 1.37\3.18 | 1.02\6.86 |

*Table 6.* Speedup over I/O-only \ compute-only across different model settings, Hardware: 2xA100, Chunk size 512, Seq-len: 16k.

| BW | 16-bit | 8-bit | 3-bit |
|------|------|------|------|
| 7Gbps | 8.10\1.10 | 4.63\1.22 | 2.39\1.63 |
| 25Gbps | 3.03\1.43 | 2.07\1.87 | 1.41\3.09 |
| 32Gbps | 2.63\1.59 | 1.85\2.12 | 1.37\3.73 |
| 56Gbps | 1.97\2.00 | 1.49\2.85 | 1.26\5.38 |

*Table 7.* Speedup over I/O-only \ compute-only under different low-precision compression, Hardware: 2xA100 100% Utilization, Chunk size 512, Seq-len: 16k, Model: Long-Alpaca-13B.

### 5.6. Incorporating KV cache Compression with *Cake*

Table 7 presents the speedup achieved by *Cake* compared to the I/O-only approach and compute-only approach across different I/O configurations and different compression ratio. We can observe similar to the effect of GQA, low-precision compression ratio reduces the KV cache size, thus enhancing the I/O load performance, thus *Cake* speedup over compute-only methods continues to increase as the precision decreases.

### 5.7. Handling Fluctuations in Resources

To evaluate *Cake*'s performance under fluctuating compute and I/O conditions, we randomly sample a compute budget trace uniformly between 0–512 tokens to represent varying computational power. Similarly, we randomly sample an I/O bandwidth trace between 0–25 Gbps to assess how *Cake* adapts its scheduling strategy.

As shown in Figure 5, *Cake* dynamically leverages both I/O and computational resources, regardless of fluctuations. Its bidirectional prefetching mechanism automatically identifies the optimal merging point to minimize TTFT, ensuring optimal performance even under varying resource constraints.

### 5.8. Evaluation on Adaptive Scheduling

In this evaluation, we give an example to evaluate the performance of the adaptive scheduling algorithm in *Cake*. We begin by sending a prefix-caching request of length 16K, representing a typical long-context prefix-caching scenario in *Cake*. To simulate a burst of incoming requests from other users, we then randomly generate 22 additional requests with sequence lengths ranging from 32 to 448, following a spiked distribution.

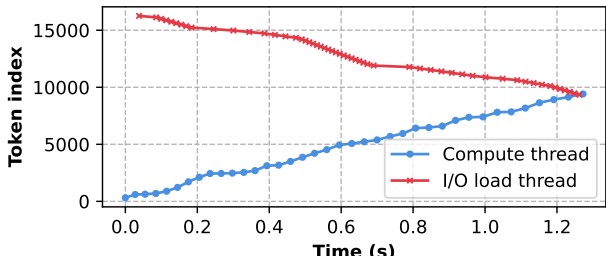

*Figure 5. Cake* trace under fluctuate network and available computation power. Hardware: A100, Model: Long-Alpaca-7B, I/O Bandwidth: 0-25Gbps, Compute Utilization: 0-512 budget, Seq-len: 16k.

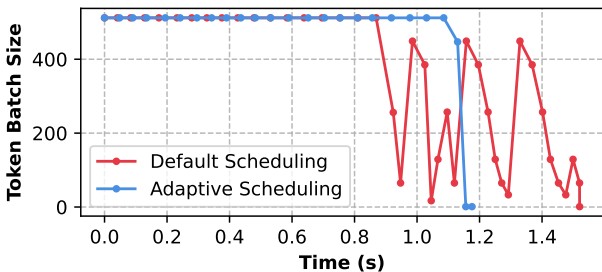

*Figure 6.* Token batch size over time under dynamic workload with default and adaptive scheduling. Adaptive scheduling of *Cake* flattens the spikes of dynamic workloads thus maximizes the system throughput.

Using these request traces, we compare the default vLLM scheduling algorithm with *Cake*'s adaptive scheduling algorithm for inference execution. The result is demonstrated in Figure 6. In vLLM's default scheduling mode, our prefix-caching request arrives first and is processed to completion before handling subsequent non-prefix-caching requests. This approach results in suboptimal GPU utilization, as the token batch size is not fully utilized at all times.

In contrast, *Cake* 's adaptive scheduling prioritizes decoding and prefill operations for incoming non-prefix-caching requests while allocating the remaining compute budget for chunk prefill of the prefix-caching request. It successfully keeps the GPU busy and reduces the overall finish time from 1.5s to 1.19s, improving throughput by 26%.

### 5.9. Overheads of *Cake*

To evaluate the overhead introduced by *Cake*, we compare the duration of each engine step between the original vLLM and vLLM with *Cake*. As shown in Figure 7, we launch a chunk prefill job on both A100 and H100 servers. The chunk prefill time of vLLM with *Cake* closely follows the trace of the original vLLM, indicating minimal performance impact.

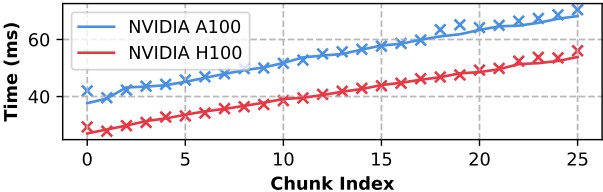

*Figure 7.* Per-step inference time in vLLM before and after integration with *Cake*. The solid line represents the step time without *Cake*, while the 'x' markers indicate step times with *Cake*.

These results demonstrate that *Cake* introduces negligible overhead, as it only performs a lightweight check to determine whether the next chunk has been fetched at runtime.

## 6. Discussion

**Compatibility with Other Acceleration Design.** Cake is orthogonal and complementary to various widely adopted acceleration methods, such as speculative decoding, multi-token prediction, KV cache quantization, eviction, and prefill-decode disaggregation. These methods can be categorized into three main classes:

*Decoding Throughput Optimization Methods*: Techniques such as speculative decoding (Leviathan et al., 2023; Li et al., 2024) and multi-token prediction (Cai et al., 2024) optimize the decoding phase by parallelizing token generation. Cake, targeting the prefill stage, is inherently compatible with these methods, allowing concurrent enhancements in both phases without conflicts.

*KV Cache Size Reduction Methods*: Approaches like KV cache quantization (Hooper et al., 2024; Kang et al., 2024; Liu et al., 2024b) and eviction (Jiang et al., 2023) reduce memory usage by compressing or selectively removing tokens from the KV cache. Cake treats the KV cache as data and is agnostic to its specific representation or compression scheme. Our evaluation with compression methods (Section 5.6) demonstrates that Cake can seamlessly integrate with these techniques, further reducing latency and enhancing performance.

*System-level Acceleration Methods*: Prefill-decode disaggregation (Zhong et al., 2024) separates prefill and decode stages onto different hardware, often using separate prefill and decode servers. Cake can be integrated into such design by managing bidirectional KV cache loading across these servers. The prefill server computes the KV cache from the beginning of the sequence and streams it forward, while the decode server concurrently manages two streams—the incoming stream from the prefill server and the reverse stream from I/O operations. The two processes meet in the middle, ensuring efficient cache preparation.

**Compatibility in Distributed Environments.** Cake's design also adapts naturally to distributed inference setups. Our evaluation with tensor-parallelism using multiple GPUs within a single node (Table 3 and Table 5) confirms that Cake effectively utilizes increased computational resources. For multi-node distributed environments, Cake operates independently within each inference engine, allowing each node to optimize its inference latency. Consequently, Cake remains effective even in extensive, distributed inference deployments.

## 7. Conclusion

In this paper, we introduced *Cake*, the first KV cache loading system that optimally balances computation and I/O to minimize Time to First Token (TTFT) in LLM inference. Unlike prior approaches that focus on either compute or I/O in isolation, *Cake* employs a bidirectional scheduling strategy that dynamically adapts to resource availability, achieving an average **2.6×** TTFT reduction compared to baselines. Additionally, its adaptive scheduling mechanism enhances system throughput, making it a practical and easily deployable solution for LLM-serving systems. Through extensive evaluations, we provide a detailed analysis of the scenarios where *Cake* is most beneficial, offering valuable insights for real-world deployment.

## Acknowledgments

We would like to thank our anonymous reviewers for their valuable comments and feedback. This work was partially supported by Boeing as well as NSF under grants CMMI-2038215, CNS-2321532, and National AI Institute for Edge Computing Leveraging Next Generation Wireless Networks, Grant–2112562.

## Impact Statement

This paper presents Cake, a system that enhances LLM inference by efficiently combining computation and I/O resources to reduce latency. By improving the responsiveness of long-context LLM applications, Cake enables smoother user experiences and broader accessibility across various domains, from conversational AI to document processing. Its efficient design can help maximize the utility of existing infrastructure, making large-scale models more practical for real-world deployment. While increased efficiency may accelerate LLM adoption, we encourage responsible deployment with fair resource allocation and transparent system usage. Overall, Cake contributes to advancing scalable and effective LLM serving for various applications.

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

# A. Details of *Cake* Algorithm

The workflow of *Cake* can be described in detail as follows:

1. Upon receiving a request, *Cake* first splits the input sequence into chunks of a predetermined size.

2. Compute the prefix hash of all the chunks and find the latest prefix hash that exists in the storage backend, and determine the $total\_tokens$.

2. Two pointers are initialized: $compute\_ptr$ starting at the beginning of the sequence (index 0), and $io\_ptr$ at the end of the sequence (index $total\_tokens - 1$).

3. Two parallel processes are initiated: a) A GPU computation thread that starts from $compute\_ptr$ and moves forward. b) An I/O streaming thread that starts from $io\_ptr$ and moves backward.

4. The GPU computation thread: - Computes KV cache for chunks starting from $compute\_ptr$. - After each chunk computation, it updates $compute\_ptr$ by adding the chunk size. - Continues until $compute\_ptr$ reaches or surpasses $io\_ptr$, or until the required KV cache is found in CPU memory.

5. The I/O streaming thread: - Fetches pre-computed KV cache for chunks ending at $io\_ptr$ from storage (local or remote) to CPU memory. - After each chunk fetch, it updates $io\_ptr$ by subtracting the chunk size. - Continues until $io\_ptr$ reaches or goes below $compute\_ptr$.

6. The process concludes when the two pointers meet or cross each other, indicating that KV cache for the entire sequence has been either computed or loaded.

7. Finally, *Cake* returns the complete KV cache for the entire sequence, ready for use in the subsequent inference steps.

This bidirectional approach allows *Cake* to efficiently utilize both computational and I/O resources simultaneously, minimizing idle time and optimizing the overall latency of KV cache preparation for long-context LLM inference.

---

**Algorithm 1** *Cake* Bidirectional KV cache Loading Algorithm

---

1: **procedure** COMPUTEKV
2:   **while** $compute\_ptr < io\_ptr$ **do**
3:     **if** ISINCPUMEMORY($compute\_ptr, COMP\_CHUNK\_SIZE$) **then**
4:       Signal I/O worker to stop
5:       **break**
6:     Compute KV cache for chunk starting at $compute\_ptr$ using GPU
7:     $compute\_ptr \leftarrow compute\_ptr + COMP\_CHUNK\_SIZE$
8: **procedure** FETCHKV
9:   **while** $compute\_ptr < io\_ptr$ **do**
10:     Fetch KV cache for chunk ending at $io\_ptr$ from storage to CPU Memory
11:     $io\_ptr \leftarrow io\_ptr - FETCH\_CHUNK\_SIZE$
12: Initialize CPU Memory, $compute\_ptr = 0$, $io\_ptr = total\_tokens - 1$
13: Start COMPUTEKV in a new thread
14: Start FETCHKV in a new thread
15: Wait for both threads to complete
16: **return** KV cache for the entire sequence

---

# B. Implementation

We implement *Cake* by extending LMCache (LMCache, 2024) and integrating it with vLLM (Kwon et al., 2023), adding approximately 1,000 lines of code.

### B.1. Enhancements to LMCache

LMCache, originally developed as the KV cache management backend for CacheGen (Liu et al., 2023), hashes token chunks into keys for efficient KV cache retrieval. To enable *Cake* to continuously receive KV cache in the background, we introduce the following enhancements:

**Asynchronous Retrieval**    We implement an asynchronous get operation to complement LMCache's existing asynchronous put functionality. This involves creating a dedicated worker thread that continuously reads chunk keys from a task queue and retrieves the corresponding KV cache to memory. Upon successful retrieval, the chunk's key is added to a resident dictionary for quick access.

**Buffer Preallocation**    We modify LMCache to preallocate chunk buffers as soon as a chunk key is pushed to the queue. This optimization allows the worker to immediately write received KV cache into memory and proceed to the next chunk without delay.

### B.2. Integration with LLM Serving Systems

*Cake* operates concurrently with LLM serving systems like vLLM. The integration process works as follows:

1. Upon receiving a request, *Cake* divides it into chunks based on the scheduled token budget.

2. Hashed keys for these chunks are pushed to the task queue in reverse order and call *batch_retrieve* API.

3. While the asynchronous get worker fetches KV cache from the end of the sequence, vLLM begins chunk prefill from the start.

4. After each vLLM engine step, *Cake* checks if the next chunk of tokens is already in the resident dictionary using the *is_resident* API.

5. If the chunk is resident, *Cake* interrupts the chunk prefill process and directs vLLM to begin token generation.

6. If the chunk is not resident, chunk prefill continues until it encounters a chunk present in the dictionary.

This bidirectional approach allows *Cake* to efficiently utilize both I/O and computational resources, potentially reducing the Time To First Token (TTFT) for long-context LLM inference tasks.

