# OpenReview forum: "Compute or Load KV Cache? Why Not Both?"
_ICML.cc/2025/Conference — ICML 2025 poster_

### Official Review · Reviewer_UkhW · 2025-03-13

**Overall Recommendation:** 4

**Summary:**

The authors proposed a simple but effective method dealing with the kv cache prefilling: utilizing both the GPU computation and IO loading  to get the free lunch of both, maximizing the loading speed with zero cost.

**Claims And Evidence:**

Yes

**Essential References Not Discussed:**

no

**Experimental Designs Or Analyses:**

it's sound enough. I like the overhead analysis, which further prove the usability of cake.
One simple concern: the authors didn't analyze the performance gains under memory-bounded and computing-bounded conditions by varying the batch size, which is a quite interesting study to fully know the potential of CAKE under different batch sizes and model sizes.

**Methods And Evaluation Criteria:**

Evaluation is mainly based on the latency metrics (wall clock time).
I don't see any inappropriate evaluation.

**Other Comments Or Suggestions:**

NA

**Other Strengths And Weaknesses:**

discussed above.

**Questions For Authors:**

NA

**Relation To Broader Scientific Literature:**

it's related to the computation efficiency of Language models, including the deployment of LLMs, VLMs, and any LM archs with a heavy prefilling overhead.

**Theoretical Claims:**

No theory claim; infra paper.

---

> ### Author Rebuttal · Authors · 2025-04-01
>
> Thank you for your interest in Cake and for providing insightful feedback. We value your constructive comments and are pleased to address your concerns.
>
> In our original experiments, we adopted the inter-token-latency-optimized configuration in vLLM v0.6.2, where the max number of batched tokens is 512. We agree that exploring different batch sizes offers valuable insight into Cake’s adaptability under memory-bound and compute-bound conditions. Below are our newly added experiment results of Cake’s performance gain over the compute-only and I/O-only prefill on different numbers of batched tokens and different model sizes. We will include such results in our revised version.
>
> -----------------------------------------------------------------
> | BatchSize | BW     | LongAlpaca-7B | LongAlpaca-13B |
> |-----------|--------|---------------|----------------|
> | 64        | 32Gbps | 1.47\3.47     | 1.47\3.55      |
> | 128       | 32Gbps | 1.83\2.40     | 1.79\2.45      |
> | 256       | 32Gbps | 2.06\2.09     | 1.96\2.20      |
> | 512       | 32Gbps | 2.20\1.94     | 2.03\1.91      |
> | 1024      | 32Gbps | 2.15\1.70     | 2.18\1.91      |
> | 2048      | 32Gbps | 2.31\1.71     | 2.04\1.70      |
> -----------------------------------------------------------------
>
> **Table r1. Speedup of Cake over I/O-only \ Compute-only method under different batch sizes of tokens and different models. Hardware: 1xA100, Seq-len: 16k**
>
> As shown in Tables r1, Cake achieves an average Time-to-First-Token(TTFT) speedup of 1.96x over I/O-only methods and 2.25x over compute-only methods across various batch sizes. At smaller batch sizes, KV cache computation is memory-bound, underutilizing compute units. As batch size increases, it shifts toward compute-bound, boosting GPU efficiency. Generally, the speedup over I/O-only methods is higher with larger batch sizes, reflecting Cake’s increasing reliance on computation. These results highlight Cake’s ability to adapt to various compute efficiencies caused by different batch sizes, automatically optimizing TTFT.
>
>
> Thanks again for your suggestion, and we are happy to address any further concerns.

---

### Official Review · Reviewer_wYCE · 2025-03-13

**Overall Recommendation:** 4

**Summary:**

The paper introduces a KV cache loading system called Cake, that optimizes computation and I/O in parallel to speed up LLM inference. It uses bidirectional scheduling for efficient resource use and adaptive scheduling to handle varying workloads. Evaluations show Cake cuts TTFT by 2.6×, making it a practical solution for long-context LLMs.

**Claims And Evidence:**

The main claim made by this paper is "Cake is the first system to demonstrate that efficiently utilizing both computational and I/O resources can optimally reduce TTFT in longcontext prefix caching scenarios."

In my opinion, this statement is clear and supported by experimmental results

**Essential References Not Discussed:**

No

**Experimental Designs Or Analyses:**

Yes, I checked the soundness of the experimental designs.

**Methods And Evaluation Criteria:**

The proposed method and evaluation criteria make sense for the problem considered.

I also really enjoy reading the analysis of the design section and the insights of Cake

**Other Comments Or Suggestions:**

I suggest introduce more details regarding chunked prefill, since Cake is built upon it. And many audience of ICML may not familar with it.

**Other Strengths And Weaknesses:**

In general, I like this paper. It is simple and intuitive

The weakness in my mind is there is no discussion on how this approach interacts with PD disaggregation, which is wildly used in practice. In my understanding, even though the KV cache is computed, they still need to be transferred to decode server through RDMA.

**Questions For Authors:**

See the weakness part.

**Relation To Broader Scientific Literature:**

TTFT is a crusial SLO in LLM serving area and scheduling long context request is a very important problem.

**Theoretical Claims:**

No theoretical claims in this paper

---

> ### Author Rebuttal · Authors · 2025-04-01
>
> Thank you for your insightful feedback and kind words about our paper. Below, we address your suggestions regarding the interaction of Cake with Prefill-Decode (PD) disaggregation. We will add the following discussion to the revised version.
> * Cake is compatible with Prefill-Decode Disaggregation with a minor modification. The typical workflow of Prefill-Decode Disaggregation involves routing a request to a prefill server to generate the KV cache, which is then streamed to a decode server. Cake can fit into this scenario by splitting its bidirectional process across these servers: the prefill server handles computation, generating the KV cache from the sequence’s start and streaming it to the decode server; On the decode server, Cake simultaneously manages two streams—one is the I/O loading stream, fetching the existing KV cache from disks starting from the last tokens, and the other is the streaming process from the prefill server, starting from the first tokens—until the two processes meet in the middle on the decode server, meaning all KV cache is ready for decoding.
> * Thank you for your suggestions to provide more details on chunked prefill. We will add more details in the background Section in our revised version.

---

### Official Review · Reviewer_opvL · 2025-03-14

**Overall Recommendation:** 4

**Summary:**

This paper introduces Cake, a hybrid KV cache loading system that leverages both I/O resources (for loading) and computational resources (for re-computation). The authors observe that both I/O-only approaches (e.g. LMCache) and compute-only approaches (e.g. vLLM) fall short in practice in terms of minimizing TTFT when loading from local or remote disks. Inspired by this, Cake uses both I/O and computation in parallel when loading from disks (which are high-capacity but low-bandwidth). Evaluations show that Cake can reduce TTFT by 2.6x on average for long-context prefix caching scenarios.

**Claims And Evidence:**

Most of the claims made by the authors are supported by citations or experiment results. Please see "Questions For Authors" for details.

**Essential References Not Discussed:**

As far as I know, there is no missing related work (in the area of KV cache loading / prefix caching systems) that remain undiscussed in this paper.

**Experimental Designs Or Analyses:**

The paper features solid and comprehensive evaluation of the proposed method, Cake.

**Methods And Evaluation Criteria:**

The paper features solid and comprehensive evaluation of the proposed method, Cake.

**Other Comments Or Suggestions:**

N/A

**Other Strengths And Weaknesses:**

N/A

**Questions For Authors:**

Thank you for submitting this paper to NeurIPS. The core idea of this paper, leveraging I/O and compute in parallel when loading KV cache from disks, is well presented and makes sense. Experiments are also solid and comprehensive.

I have two major questions/concerns.
1. In reality, I/O (network) and compute capability can vary dramatically from time to time, even with the same set of local/remote hardware, depending on if you have competing workloads, network variance, temperature of hardware, etc. Is your system assuming a fixed config of I/O and compute capability, pre-measured given the hardware stack? If this is the case, do you have real-time profiling that adjusts the capabilities so that your scheduling works well for the real-world scenario? Otherwise it seems very likely that sub-optimal scheduling decisions will occur.
2. Do you think your system would work for CPU->GPU loading with minor adaptation? If not, what do you think is the major challenge (or maybe loading is all you need in that case)? Is it because disk->CPU/GPU might be the only case where re-computation can be sometimes more efficient (in terms of TTFT) than direct loading?

**Relation To Broader Scientific Literature:**

This paper lies within the broad area of KV cache management systems for language model inference. The authors focus primarily on acceleration of KV cache loading from disks, which can be helpful to cloud providers in the case of prefix-cache loading or prompt-cache loading.

**Theoretical Claims:**

N/A

---

> ### Author Rebuttal · Authors · 2025-04-01
>
> Thanks for your insightful feedback, and we are glad to address your concerns.
> 1. **Dynamic I/O, workloads, etc.** Cake’s parameter-free design allows it to automatically adapt to dynamic conditions, such as fluctuating network bandwidth or GPU performance, as discussed in Part 3 of Section 4. This is a key strength, as shown in Figure 5 of Section 5.6, we conducted an experiment on dynamic computational power and network bandwidth. Under severe fluctuations, Cake is still able to find the optimal merging point automatically and maximize the utilization of dynamic resources. The reason behind it is that Cake does not rely on any offline profiling or any static parameters. The optimal utilization of compute and I/O is achieved by our bidirectional KV cache generation mechanism, demonstrated in Figure 2. Cake starts KV cache computing and loading from opposite directions, chunk by chunk; the two threads run in parallel asynchronously and merge at a certain point in the middle of the context. A sudden network traffic burst may slow down the I/O transfer, then the two threads will merge at a position closer to the end of the context. Similarly, if a temperature spike reduces GPU frequency and slows down the computing progress, the threads will meet at a position closer to the head of the context. The scheduler doesn’t need to change anything caused by a certain period of exceptional condition, it only needs to interrupt the two threads if the last computed chunk includes a loaded token‘s cache and switch request status to the decoding stage. This design ensures that Cake always achieves the optimal Time-To-First-Token as it fully leverages the available I/O and computing capability, even though they are dynamic.
> 2. **Potential application of CPU-GPU loading.** Cake can be applied between CPU and GPU with minor adjustments. Yet, there are two reasons why we don’t suggest applying Cake between host memory and GPU. First, as we show in Figure 3, for a context length of 32k, the KV cache computing speed of an H100 is approximately equivalent to 4GB/s I/O transfer. However, H100 supports PCIe 5.0x16, which provides up to 64GB/s host-to-device transfer bandwidth per GPU. This huge gap makes re-computing less efficient, and loading all the cache is a better option. Second, compared with SSD and remote disks, host memory is more expensive, and its storage is limited. As is profiled by AttentionScore, 80% of cache hits happen on the disk level, and we believe optimizing it will bring broader benefits to the modern LLM serving system.
>
> We appreciate your interest in Cake and your insightful reviews. We are happy to address any further questions about Cake.

---

> > ### Comment · Reviewer_opvL · 2025-04-08
> >
> > I appreciate the authors for their detailed and helpful response. I have also read all other reviewers' comments in detail.
> >
> > Overall, I like the research direction this work opens (leveraging both I/O + compute for efficient KV cache management). I can foresee related ideas like I/O+compute for loading from heterogeneous storage, e.g. CXL memory, serverless storage, etc., or better policy design under the framework of Cake. It might require some additional effort if you're sending something related to OSDI/SOSP though.
> >
> > I decide to raise my score to 4. The main reason is because (1) my concerns are addressed, and (2) the authors actually integrate Cake into LMCache and vLLM --- So hopefully the impact of the idea will go beyond the paper itself.

---

> > > ### Author Response · Authors · 2025-04-08
> > >
> > > Thank you for your kind words and thoughtful engagement with our work. We’re excited that you see potential in Cake’s direction and future applications, and we sincerely appreciate your feedback throughout this process.

---

### Official Review · Reviewer_bg9y · 2025-03-18

**Overall Recommendation:** 3

**Summary:**

This paper introduces Cake, a novel KV cache loading system that optimally utilizes both computational and I/O resources in parallel, with a bidirectional scheduling strategy, and an adaptive scheduling mechanism. The proposed method can be seamlessly integrated in existing methods, with better TTFT.

**Claims And Evidence:**

Most of the claims and evidence in the paper are convincing.

**Essential References Not Discussed:**

N/A

**Experimental Designs Or Analyses:**

1. The experiment designs of this paper are good.
2. It would be better if the authors conduct experiments on more mobile devices, such as GPU for laptops (4090, 3090) and GPU for mobile phones.

**Methods And Evaluation Criteria:**

Most of the evaluation criterions are good.

**Other Comments Or Suggestions:**

N/A

**Other Strengths And Weaknesses:**

1. The most significant strength of this paper is that the proposed method does not bring any harm to model accuracy, making it useful in most applications.
2. Please discuss whether the proposed method conflicts with other acceleration methods for KV cache and decoding, such as speculative decoding, multi-token prediction, KV cache quantization, and eviction. Intuitively, if these methods are applied, the improvements of Cake may be limited.
3. Please discuss whether the proposed method is compatible with other system-level acceleration methods, especially for the methods that separate the prefilling and decoding stages in different hardware.
4. Please discuss the performance of Cake when multiple GPUs and even multiple GPU nodes are employed. Does distributed inference impacts Cake?
4. Typos: There is no ``.'' in table 4 and table 2.

**Questions For Authors:**

N/A

**Relation To Broader Scientific Literature:**

I did not find any specific relation between this paper and broader scientific literature.

**Theoretical Claims:**

There is no theoretical claims in the paper.

---

> ### Author Rebuttal · Authors · 2025-04-01
>
> Thank you for your insightful comments and feedback on our paper. We are pleased to address your concerns regarding the compatibility and performance of our proposed method, Cake, with other acceleration techniques and system-level optimizations. We will add the following discussion to the revised version.
>
> * **Compatibility with other acceleration methods.** Thanks to the simple and elegant design, Cake is orthogonal and complementary to the broad acceleration methods you mentioned, including speculative decoding, multi-token prediction, KV cache quantization, eviction, and Prefill-Decode disaggregation, making Cake very flexible to deploy. These methods can be categorized into three main classes:
> * + **Techniques to Accelerate LLM Decoding Throughput**:
> * + + Methods like speculative decoding and multi-token prediction are focused on optimizing the decoding stage, e.g., by leveraging parallel generation opportunities to address the inefficiencies of autoregressive decoding.
> * + + Cake, however, targets the prefill stage, i.e., generating the KV cache for the decode stage. Since the prefill and decoding stages are distinct, there is no conflict between Cake and these decoding-focused methods. They can be used together to enhance overall inference efficiency.
> * + **KV Cache Size Reduction Methods**:
> * + + Techniques such as quantization and eviction aim to reduce the memory footprint of the KV cache. Cake regards the KV cache as simple data, and it does not restrict the size of KV  cache or how the KV cache is computed. Therefore, these methods can be seamlessly integrated with Cake.
> * + + Actually, we discussed the integration of KV cache compression methods in Section 5.5 of our paper, where we evaluated Cake’s performance with compression techniques, showing that they are orthogonal and can complement Cake’s bidirectional scheduling strategy. This synergy could lead to additional reductions in TTFT, though the extent of improvement may vary depending on specific implementations and resource constraints.
> * + **System-level Acceleration Methods**:
>
> * + + **Prefill-Decode Disaggregation.** Cake is compatible with Prefill-Decode Disaggregation with a minor modification. The typical workflow of Prefill-Decode Disaggregation involves routing a request to a prefill server to generate the KV cache, which is then streamed to a decode server. Cake can fit into this scenario by splitting its bidirectional process across these servers: the prefill server handles computation, generating the KV cache from the sequence’s start and streaming it to the decode server; On the decode server, Cake simultaneously manages two streams—one is the I/O loading stream, fetching the existing KV cache from disks starting from the last tokens, and the other is the streaming process from the prefill server, starting from the first tokens—until the two processes meet in the middle on the decode server, meaning all KV cache is ready for decoding.
>
> * + + **Distributed inference.** We have evaluated Cake’s performance with tensor-parallelism using two GPUs, as detailed in Table 3 and Table 4 of our paper. The results demonstrate that Cake can effectively utilize the increased computational capabilities provided by multiple GPUs within a node. By distributing the model across GPUs, the computational power available for Cake’s bidirectional scheduling improves, allowing it to optimize TTFT more efficiently. For distributed inference settings, where multiple inference engines are deployed across various nodes in the data center and requests are routed to different engines for processing, Cake can still be applied within each inference engine. Each engine can independently optimize its own inference latency using Cake, ensuring that the method remains effective even in large-scale, distributed environments.
>
> * **About mobile devices**: Cake is designated for long-context prefix-caching enabled LLM inference systems and is beneficial as long as the simultaneous KV computation and I/O KV cache loading is possible. The enterprise-level data center is the best use case because of its sufficient GPU computational power, GPU memory capacity, and I/O bandwidth. The idea of Cake could also be applied to mobile devices, but the limited computation power and memory space limit the model size or context length, making mobile computing not the ideal use case. Cake’s deployment on mobile devices is promising but needs future work to address engineering challenges.
>
> * Thank you for your careful reading and pointing out the typos. We will correct them in the final version to ensure clarity.

---

### Decision · Program_Chairs · 2025-05-01

**Decision:**

Accept (poster)

**Comment:**

KV fetching becomes an ever more important problem as models get longer context, and as ChatGPT-like systems become even more widely adopted. This paper proposes Cake, a KV-cache loading system that uses a clever scheduling strategy to enable more efficient loading of the KV cache. The reviewers generally agreed that the proposed approach is sensible, solves a practical problem, and backed up by solid experiments. A clear accept.